# Geo-Epidemiology of Malaria at the Health Area Level, Dire Health District, Mali, 2013–2017

**DOI:** 10.3390/ijerph17113982

**Published:** 2020-06-04

**Authors:** Mady Cissoko, Issaka Sagara, Moussa H. Sankaré, Sokhna Dieng, Abdoulaye Guindo, Zoumana Doumbia, Balam Allasseini, Diahara Traore, Seydou Fomba, Marc Karim Bendiane, Jordi Landier, Nadine Dessay, Jean Gaudart

**Affiliations:** 1Malaria Research and Training Center—Ogobara K. Doumbo (MRTC-OKD), FMOS-FAPH, Mali-NIAID-ICER, Université des Sciences, des Techniques et des Technologies de Bamako, Bamako 1805, Mali; isagara@icermali.org (I.S.); aguindo15@yahoo.fr (A.G.); jean.gaudart@univ-amu.fr (J.G.); 2Aix Marseille Université (AMU), Institut national de la santé et de la recherche médicale (INSERM), Institut de Recherche pour le Développement (IRD), 13005 Marseille, France; melmadine@hotmail.com (S.D.); marc-karim.bendiane@inserm.fr (M.K.B.); jordi.landier@ird.fr (J.L.); 3Direction Régionale de la Santé de Tombouctou, Tombouctou 59, Mali; tondifarma1@yahoo.fr (M.H.S.); zoumana.doumbia@yahoo.fr (Z.D.); allousmed@yahoo.fr (B.A.); 4Mère et Enfant face aux Infections Tropicales (MERIT), IRD, Université Paris 5, 75006 Paris, France; 5Programme National de la Lutte contre le Paludisme (PNLP Mali), Bamako 233, Mali; dtkone@hotmail.fr (D.T.); drfomba@hotmail.fr (S.F.); 6ESPACE-DEV, UMR228 IRD/UM/UR/UG/UA, Institut de Recherche pour le Développement (IRD), 34093 Montpellier, France; nadine.dessay@ird.fr; 7Aix Marseille Université, APHM, INSERM, IRD, SESSTIM, Hop Timone, BioSTIC, Biostatistic & ICT, 13005 Marseille, France

**Keywords:** malaria, hotspot and incidence, spatio-temporal analysis, environment and meteorology, geo-epidemiology

## Abstract

*Background*: According to the World Health Organization, there were more than 228 million cases of malaria globally in 2018, with 93% of cases occurring in Africa; in Mali, a 13% increase in the number of cases was observed between 2015 and 2018; this study aimed to evaluate the impact of meteorological and environmental factors on the geo-epidemiology of malaria in the health district of Dire, Mali. *Methods*: Meteorological and environmental variables were synthesized using principal component analysis and multiple correspondence analysis, the relationship between malaria incidence and synthetic indicators was determined using a multivariate general additive model; hotspots were detected by SaTScan. *Results*: Malaria incidence showed high inter and intra-annual variability; the period of high transmission lasted from September to February; health areas characterized by proximity to the river, propensity for flooding and high agricultural yield were the most at risk, with an incidence rate ratio of 2.21 with confidence intervals (95% CI: 1.85–2.58); malaria incidence in Dire declined from 120 to 20 cases per 10,000 person-weeks between 2013 and 2017. *Conclusion*: The identification of areas and periods of high transmission can help improve malaria control strategies.

## 1. Introduction

According to World Health Organization (WHO) estimates, malaria is the leading cause of death in children under five years of age, with a mortality rate of 22% [1]. In recent decades, the global fight against malaria has made a lot of progress with encouraging results, despite disparities among countries [1]. In Africa, vector control and access to care (screening and rapid treatment) helped reduce the number of clinical cases by 40% between 2000 and 2014 (i.e., 663 million cases were avoided) [2]. Over the same period, attributable mortality was reduced by 60% [3]. Since 2014, however, malaria incidence has remained stable, with a slight increase between 2016 and 2017 [3]. Malaria cases in Mali account for 3% of the malaria burden in Sub-Saharan Africa. The number of confirmed cases in the country increased from 1,520,047 in 2015 to 1757,292 in 2018, as reported by the National Malaria Control Program. The number of malaria cases in Mali accounts for 3% of the malaria burden in Sub-Saharan Africa. The factors associated with this increase in malaria incidence need to be identified and addressed in malaria control strategies.

The malaria elimination goals set by WHO for the year 2030 are to reduce the incidence of cases and mortality by at least 90%, to eradicate malaria in at least 35 countries, and to prevent re-infection in malaria-free countries. These goals are to be achieved through early diagnostic and treatment, targeted chemoprevention, vector control and epidemiological surveillance [4]. This strategy has already resulted in an important reduction of malaria cases in Senegal, a Sahelian country that neighbors Mali and that also has a highly seasonal epidemiology [5]. In spite of this, resources allocated for malaria control in the Sahel remain below projected needs [4], and were even reduced during the 2017–2018 period [3,6].

Mali is characterized by five ecoclimatic zones: the Sahara zone, the Sahelian zone, the Sudanese zone, the Sudanese-Guinean zone, and the inner delta of the Niger river [7,8]. Malaria transmission in Mali is both endemic and epidemic [7], with *Plasmodium falciparum* being the most common parasite. However, the distribution of malaria transmission risk is heterogeneous between the southern and the northern regions of the country [7,8]. In the south, transmission is hyperendemic or holoendemic, with a prevalence of *Plasmodium falciparum* [7] in children aged 2–10 years ranging from 10% to 50% (PAPfPR2-10, 10%–50%) [7]. By contrast, in the north, transmission is hypoendemic, with a prevalence of *Plasmodium falciparum* estimated at <10% (PAPfPR2-10, <10%) [7] and a growing presence of *Plasmodium vivax* (the parasite now accounts for 40% of cases) [9].

Increasing desertification, which is accompanied by lower rainfall, higher temperature, and sparser vegetation, may partly explain the lower risk of malaria transmission in the north. In addition, the contrasting meteorological and environmental characteristics of northern and southern regions result in different type of agriculture development, which may in turn play a role in the heterogeneity of malaria transmission risk in the country. Unfortunately, in spite of these socio-environmental variations, the national malaria control program applies similar strategies during identical periods.

Studies have well described the relationship between malaria incidence and meteorological factors, which is characterized by a lag ranging from one to three months [10]. The relationship between malaria incidence and environmental factors like land use and land cover is more controversial, particularly with regard to agricultural land use. The development of agriculture has been found to be associated with a decrease in malaria incidence in some deforested areas and with an increase in mosquito populations and malaria transmission in semi-desert areas [11,12]. The question of land use and land cover is especially important in the Sahel, where efforts to adapt to climate change and to improve food security have been accompanied by agricultural land use programs. Indeed, while one might have expected the number of mosquito breeding sites to diminish in the region as a result of recent droughts, the development of artificial ponds and irrigated areas has in fact caused an increase in the number of these sites [12].

In Mali, the impact of meteorology on the geo-epidemiology of malaria has been widely studied, and that of land use and land cover has recently received some attention. However, available studies [13,14,15,16] may not reflect the overall situation in the country, and this for two reasons. First, meteorological, and environmental characteristics vary significantly across the country. Second, environmental change caused by natural and human factors (climate change, agricultural development, human mobility) may be significantly modifying the risk of malaria transmission, in particular via changes in vector species composition and vector population size [17,18]. In this context, it seems important to determine the contemporary dynamics of malaria transmission in Mali at both the local and regional scales.

With this in mind, we set out to explore the geo-epidemiology of malaria in the health district of Dire, northern Mali, with a simultaneous focus on meteorological factors and on the specific environmental factors of land use and land cover. This region, which is representative of the Sahel, is of particular interest in that it is inhabited by sedentary populations that are increasingly concentrated in arable areas [19], where they practice irrigated cultivation with the help of seasonal workers [20,21]. As such, our work updates the results of studies from the 1990s that described malaria transmission in Dire as seasonal and stable over time, with a spatially heterogeneous distribution linked to the presence of ponds and the Niger River [22,23].

This study aimed to evaluate the impact of meteorological and environmental factors on the spatio-temporal distribution of malaria in the health district of Dire, in Sahelian Mali, with a view to developing malaria control strategies that target areas and periods with a high risk of malaria transmission.

## 2. Materials and Methods

### 2.1. Location of the Study

The study was conducted in Dire, one of the five health districts that make up the Timbuktu region. According to updated data from the General Census of Population and Housing of Mali (RGPH-2009), the population of Dire in 2017 was 143,219 inhabitants for an area of 1750 km^2^, corresponding to a density of approximately 50 inhabitants/km^2^. Dire is located in the inner delta of the Niger River, which is flooded half of the year (from August to February) [21]. During this period, populations are isolated, and the few means of transport in use (pinnasses and pirogues) are rarely available. The climate is Sahelian, with large temperature differences between day and night, an average temperature of 33 °C (minimum 21 °C and maximum 45 °C), and an average annual rainfall of 230 mm (180 to 430 mm) [21].

### 2.2. Data Collection and Sources

The health system of Mali has a pyramidal structure, with operational organizations at the base that offer primary health care and that correspond to health areas. The latter are grouped together into health districts, each with its own district hospital. The health district of Dire is composed of 18 health areas.

#### 2.2.1. Confirmed Cases and Population

Data on malaria cases (confirmed by rapid diagnostic tests) were obtained from weekly reports submitted to the Dire health district by the 18 health areas, as requested by the National System of Health and Social Information (SNISS). Data on population in the different health areas were drawn from the updated RGPH-2009. Ethical Approval and Consent to Participate: The study was approved by the Ethics Committee of the Faculty of Medicine and Odonto-stomatology of Bamako n° 2017/05/CE/FMPOS. No individual information was collected or analyzed in the course of this work. Only aggregated, collective data were analyzed.

#### 2.2.2. Meteorological and Environmental Data

Meteorological data on river height (cm), rainfall (mm), and the number of rainy days were obtained from the local agricultural service. They were completed with data on temperature °C, source: Modern-Era Retrospective analysis for Research and Applications Version 2 (GMAO, MERRA-2, Greenbelt, MD, USA), resolution 0.5 × 0.625°, relative humidity %, source: Atmospheric Infrared Sounder (JPL, AIRS, Pasadena, CA, USA) resolution 1°, wind speed (km/h), and rainfall (mm) [24]. Weekly rainfall per health area (mm, source: IMERG-5 resolution 0.1°) was determined based on daily remote sensing data [24].

Environmental data on land use (agricultural yield and cultivated land area) were obtained from the local agricultural service. They were completed with data on land cover expressed by the normalized difference vegetation index, or NDVI source: Moderate-Resolution Imaging Spectroradiometer Terra (MODIS-Terra) resolution 0.05°). The following data on the health areas were collected as binary variables through field study: off-season cultivation, proximity to the river, propensity for flooding, and presence of lowlands (Appendix A). The geographical coordinates of the health areas were obtained from the National Directorate of Statistics and Informatics and were completed using mobile Global Positioning system (GPS) tracking.

### 2.3. Statistical Data Analysis

#### 2.3.1. Temporal Analysis

For the temporal analysis, the following data were aggregated per week using means [25], quartiles, and variations: (i) temperature: maximum, minimum, median; (ii) relative humidity: day humidity and night humidity; (iii) river height; (iv) rainfall; and (v) bathymetry. Weekly disaggregation of monthly NDVI data [23] was performed using linear interpolation [26,27].

The relationship between malaria incidence and meteorological and environmental factors was investigated. To avoid collinearities, the size of the meteorological and environmental dataset was reduced using principal component analysis [28]. Three main components were determined with the elbow method and the Kaiser criterion [28]. These main components were used as synthetic indicators (SIs) whose impact on malaria incidence was determined with a generalized additive model (GAM) [29]. A negative binomial distribution was conducted to adjust for over-dispersion [30]. The population was log transformed to obtain incidence rate ratios (IRRs). Spline smoothing was performed to study the non-linear relationship between malaria incidence and SIs [29].

“The following model was used:
(1)$$\begin{array}{cc} log\left(Cases\left(T\right)\right)= & log\left(Population\left(T\right)\right)+f_{1}\left(ISI_{1}\left(T-lag_{1}\right)\right)+f_{2}\left(ISI_{2}\left(T-lag_{2}\right)\right)\\ & +f_{3}\left(ISI_{3}\left(T-lag_{3}\right)\right)+\varepsilon \end{array}$$
where *T* was the study period; *Population*, the population during *T*; *Cases*, the total number of malaria cases during *T*; *I**S1*, *IS2*, and *IS3* synthetic indicators of the environmental and meteorological variables yielded by the PCA; *f1*, *f2*, and *f3:* spline functions; *ε* the residuals whose covariance matrix had a first-order autoregressive structure; lag1, lag2, and lag3, lags determined by univariate GAM analysis, based on the minimization of generalized cross-validation (GCV) and the maximization of the explained deviance [10,29].

#### 2.3.2. Correction of Recording Bias

The malaria cases recorded in hospital accounted for 19% of all malaria cases. Given the large number of cases and the fact that we lacked information to link the cases to the health areas, we corrected the recording bias to avoid power loss. The malaria cases recorded in hospital were distributed among the different health areas using an impedance model [31]. The impedance model allowed to estimate the mobility flows between localities, based on distance and population density.

#### 2.3.3. Spatial Analysis

The evolution of spatial heterogeneity over the study period was analyzed by breaking down the time series of malaria incidence according to rainy season and dry season (nine time periods) and by accounting for the incidence lags identified in the temporal analysis.

A search for malaria hotspots and coldspots (i.e., high and low risk health areas) was performed with SaTScan for each time period [32,33]. An elliptical window was centered on the different health areas with a radius ranging from 1 to 50% of the population [33]. The Oliveira test was performed to account for the edge effects of the detected hotspots and coldspots [34]. Malaria incidence was mapped based on this spatio-temporal information, and the location of high and low risk health areas by time period was added to the generated maps. Shapefiles were extracted from the GADM (version 3.6, Davis, CA, USA) Center for Spatial Sciences at the University of California, Davis and Open Street Map [35] websites.

#### 2.3.4. Meteorological and Environmental Factors Associated with Hotspots

Multiple correspondence analysis was performed to study the relationship between malaria hotspots and selected meteorological and environmental factors at the health area level. The analyzed variables were: rainfall, agricultural yield, cultivated land area, proximity to the river, propensity for flooding, presence of lowlands, population size, and urbanization. We used means, medians, and totals measured in each health area over the study period for quantitative variables, and then transformed the latter into categorical variables. An unsupervised hierarchical classification was applied to 10 components that explained approximately 99% of inertia [28,36]. The risk classes thus obtained corresponded to the meteorological and environmental profiles of health areas. The link between the meteorological and environmental profile of a health area and the number of time periods during which this health area was a hotspot, was estimated using univariate and then multivariate analysis. Univariate analysis was performed using (non-parametric) Fisher’s exact test and Kruskal–Wallis test. Multivariate analysis was performed with a GAM model. A bivariate spline function of geographic coordinates was used to adjust for the purely spatial effect. A negative binomial distribution was used to adjust for over-dispersion of the total number of malaria cases during the study period (dependent variable). The risk classes were used as independent variables. The population was log transformed to obtain the IRR associated with each meteorological and environmental profile. The various statistical tests were performed with an α = 0.05.

### 2.4. Software and Packages

The various statistical analyses were performed using R software version 3.4 (R Development Core Team, R Foundation for Statistical Computing, Vienna, Austria) packages {mgcv}{caschrono}{FactoMineR}{forecast}{ggplot} [36,37]. The QGIS software version 3.10.0 (Open Source Geospatial Foundation Project, Beaverton, OR, USA) was used for mapping [38] and the SaTScan software version 9.6 (Information Management Services Inc., Calverton, MD, USA) for identifying clusters [33]. Painet.net version 4.10 (Warren Paint and Color C., Nashville, TN, USA) was used for image processing.

## 3. Results

### 3.1. Temporal Analysis

Between 2013 and 2017, the time series of malaria incidence showed high inter- and intra-annual variability (Figure 1). Periods of high transmission started in August or September and ended in February or March, depending on the year. Malaria incidence decreased from 43 per 10,000 person-weeks in 2013 to less than 20 per 10,000 person-weeks in 2017 (Table 1 and Appendix A).

Principal component analysis of the meteorological and environmental variables yielded 3 SIs [36] that explained 73.7% of inertia (Appendix A). The first SI (43.2% of inertia) consisted of temperature vs. river height. The second SI (19.2% of inertia) was composed of rainfall, relative humidity, bathymetry, and NDVI. The third SI (11.5% of inertia) consisted mainly of wind speed. The univariate analysis performed with the GAM model identified the lags between meteorological and environmental factors and malaria incidence. SI 1 (temperature vs. river height) showed a lag of seven weeks, SI 2 (rainfall, relative humidity, bathymetry, and NDVI) a lag of 11 weeks, and SI 3 (wind speed) a lag of four weeks (Appendix A).

The multivariate analysis performed using the GAM took these lags into account.

The relationship between SI 1 (temperature vs. river height) and malaria incidence (*p* < 0.001) was significant after a lag of 7 weeks (Figure 2a): the impact of temperature was positive at first and then negative with high values, while the impact of river height was negative at first and then positive with high values. The relationship between SI 2 (rainfall, relative humidity, bathymetry, and NDVI) and malaria incidence was significant after a lag of 11 weeks (Figure 2b) (*p* < 0.001): the impact of all variables was positive at first and then negative with high values. SI3 (wind speed) was not significantly associated with malaria incidence.

### 3.2. Spatial Analysis of Malaria Incidence by Satscan

The central health areas were at low risk of malaria all year round, the northern health areas were at high risk all year round, and the western health areas were at high risk during the rainy season. Incidence varied by time period in all health areas. The health areas of Koura, Kirchamba, Gari, and Dangha were hotspots during almost all time periods, with incidences of 133, 65, 104, and 82 per 10,000 person-weeks, respectively. During the rainy period, the risk tended to increase in the health areas of Kondi, Alwalidji, and Issafaye (110, 86, 179 per 10,000 person-weeks, respectively). Relative risk varied between 1.73 to 2.66 in hotspots and between 0.23 to 0.43 in coldspots. These results are presented in Figure 3 and Appendix A. Rainfall was irregular and river height increased over the study period (Table 1).

### 3.3. Relationship between Meteorological and Environmental Factors and Hotspots

In order to identify the meteorological and environmental factors associated with the risk of a health area being a hotspot, we synthesized the meteorological and environmental indicators using multiple correspondence analysis. The unsupervised hierarchical classification performed on the first 10 components of the multiple correspondence analysis (99.99% of inertia) yielded three risk classes (Appendix A). Class 1 (Garbacoira, Dangha, Koura, Kirchamba, Gari, BSA, and Arham) was composed of health areas characterized by proximity to the river, low median rainfall, high agricultural yield, and propensity for flooding. Class 2 (Kondi, Issafaye, Salakoira, Alwalidji, Kabaika, and Chirfiga) was made up of health areas characterized by proximity to the river, intermediate median rainfall, intermediate agricultural yield, and propensity for flooding. Lastly, Class 3 (Sarayamou, Dire, Haibongo, and Tindirma) was composed of health areas characterized by distance from the river, high median rainfall, low agricultural yield, and low flooding. These results are presented in Figure 4 and Figure 5.

The univariate analysis performed with Fisher’s exact test and Kruskal–Wallis test identified the meteorological and environmental variables associated with the risk of a health area being a hotspot. High agricultural yield (Figure 4b), low median rainfall (Figure 4c), propensity for flooding (Figure 4e), off-season cultivation (Figure 4f), presence of lowlands (Figure 4g), and proximity to the river (Figure 4h) were significantly associated with a high number of time periods during which a health area was a hotspot, with *p*-value respectively at 0.04, 0.035, 0.01, 0.003, 0.049, and 0.002. On the other hand, urbanization (*p*-value = 0.091) and large population size (*p* value = 0.079) were not significantly associated with the risk of a health area being a hotspot. The association between risk classes and the risk of health area being a hotspot was significant, with *p*-value at 0.01 (Figure 4a).

Multivariate analysis determined the IRR of the different risk classes using Class 3 (health areas characterized by distance from the river, high median rainfall, low agricultural yield, and low flooding) as a reference. Class 1 (health areas characterized by proximity to the river, low median rainfall, high agricultural yield, and propensity for flooding) had an IRR of 2.21 (95% CI: 1.85–2.58) and Class 2 (health areas characterized by proximity to the river, intermediate median rainfall, intermediate agricultural yield, and propensity for flooding) had an IRR of 1.78 (95% CI: 1.53–2.03). The regressive model showed an adjusted R-squared of 80.3% (Table 2).

## 4. Discussion

Our study conducted in Dire, Mali, found high inter- and intra-annual variability of malaria incidence. The seasonal distribution of transmission was bimodal, with a first peak from August to September and a second one from December to February and even to March. As elsewhere in the Sahel [23], the period of high malaria incidence in Dire generally straddled two administrative years, requiring us to perform analyses by transmission period as opposed to calendar year. The two peaks were modulated by rainfall and river height, respectively, and the peak linked to river height grew wider with time. This period also coincided with the arrival of seasonal workers [39]. Older studies had found the epidemiology of malaria in Dire to be seasonal and stable over time. Our divergent findings may be the result of a combination of climate change and changes in population flows [23]. Temporal variability of malaria incidence due to the influx of seasonal workers has been observed in South Africa [40] and in some health districts of Madagascar [41,42,43].

A gradual reduction in malaria incidence was also observed over the study period (Appendix A). This may be explained by the implementation of malaria control strategies in the region, including seasonal chemoprevention and mosquito net distribution, both of which covered approximately 99% of the Dire health district.

Our study found a linear relationship between malaria incidence and SI 1 (temperature vs. river height). This result is similar to findings reported in other studies [44,45]. The relationship between malaria incidence and SI 2 (rainfall, relative humidity, bathymetry, and NDVI) was non-linear, with a lag of 11 weeks, which may be explained by the bathymetry effect and by the fact that larval sites are washed out by the rain. Also in Mali, Coulibaly [44], Gaudart [13] and Sissoko [45] found a non-linear relationship between malaria incidence and rainfall, relative humidity, and NDVI, with a similar lag of three months [45].

According to our spatial analysis, health areas characterized by high median rainfall and by distance from the river had a lower risk of malaria transmission [46,47]. Malaria transmission was shown to persist in all health areas [48]. The risk of transmission increased over time in the health areas of Koura, Kirchamba, Issafaye, Dangha, and Alwalidji (Hierarchical 3 and Appendix A) [14], all of which were characterized by proximity to the river, high agricultural yield, the inflow of seasonal agricultural, and fishing workers during the rainy season (Figure 5). Studies conducted in Mali [9], Ivory Coast [49], and Gambia [50], as well as a laboratory trial carried out in South Africa [51] have found a similar association between malaria and intensive land use, in particular irrigated cultivation and off-season cultivation (market gardening) [52,53,54,55]. This association may be explained by the fact that intensive land use leads to a rise in pond water levels, not only due to the large amounts of water it requires but also to the soil erosion it induces, which in turn causes water stagnation in ponds. It should be noted that this phenomenon has been on the increase in the Sahel region despite global warming [37].

Our study has limitations that must be acknowledged. First, our analyses were based on health data obtained from public services (i.e., the number of clinical malaria cases, confirmed by rapid diagnostic tests). Such data usually come with a recording bias: on the one hand, the district hospital attracts mainly nearby communities, and, on the other hand, villages located within a health facility catchment area are more likely to report cases the closer they are to the health facility [56]. This recording bias was corrected using a mobility model [31] to account for the probable origin of cases, and all our analyses were performed using meteorological and environmental data collected at the same scale as malaria cases (health area). Given ongoing improvements in health information in Mali, in other Sahelian countries, such data are increasingly being used for research and decision-making purposes. In fact, our methodology could be used by other local health authorities to determine the geo-epidemiology of malaria at the health area level. Second, available datasets we used to characterize meteorological and environmental variables. One of these datasets was obtained through remote sensing, and, consequently, was not specifically designed for our study region. In order to compensate for this, we combined data from different sources (data on agricultural yield obtained from the local agricultural service, rainfall data obtained through remote sensing) and we identified the factors that remained consistent across datasets.

## 5. Conclusions

The characteristics of the Dire health district, which are typical of the Sahel region, should be taken into consideration in the development of malaria control strategies. As our findings indicate, seasonal chemoprevention in Dire should be implemented until February or March, instead of November as is currently the norm. Health areas at greater risk of transmission, are characterized by proximity to the river, propensity for flooding, and high agricultural yield. These health areas could benefit from specific control strategies during high-transmission periods: community interventions to facilitate rapid diagnosis and trea Hierarchical classification tment, intensification of education and awareness raising activities, additional distribution of mosquito nets, and mass treatment to reduce asymptomatic carriage. Future studies should examine the relationship between malaria incidence and socio-economic, meteorological, and environmental factors at the village/concession scale and at the regional/country scale. Given that climate change, economic development, and public health interventions are likely to rapidly transform the dynamics of malaria transmission, the evolving geo-epidemiology of malaria in the Sahel region needs to be properly understood to guarantee the success of malaria control strategies.

## Figures and Tables

**Figure 1 ijerph-17-03982-f001:**
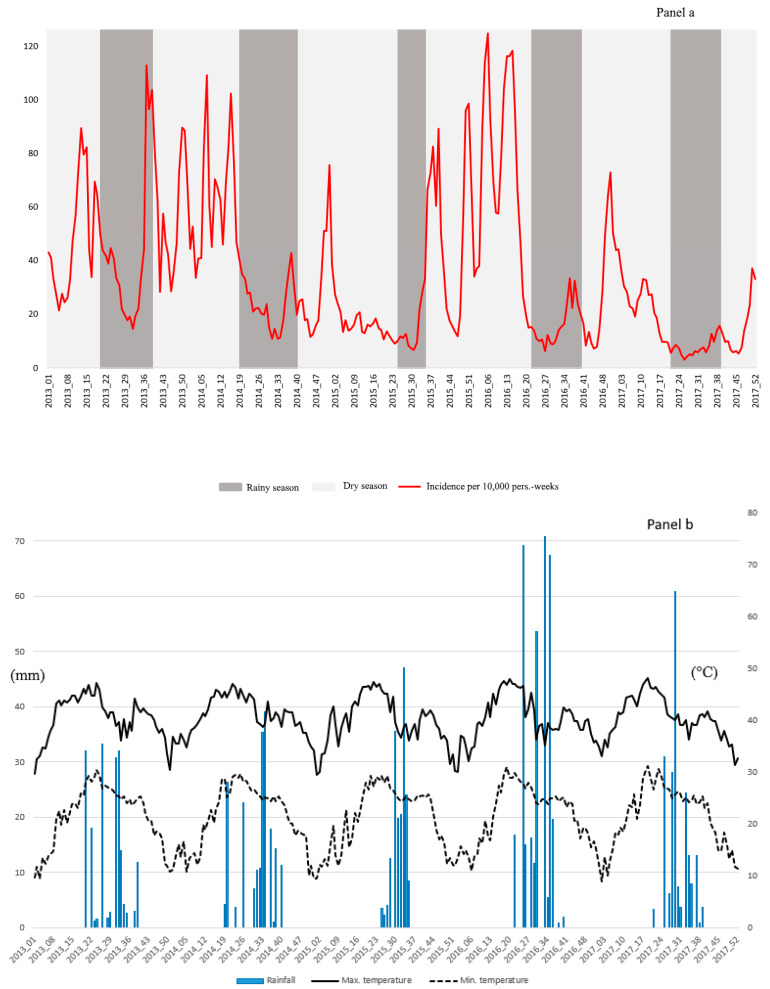
Evolution of weekly malaria incidence according to meteorological factors in Dire health district, 2013–2017. (**a**) The incidence per 10,000 person-weeks is represented by the red line, the dry season by the light grey bar, and the rainy season by the dark grey bar. (**b**) Weekly rainfall accumulation (mm) is represented by the dark blue histogram, median maximum temperature (°C) by the solid black line, and median minimum temperature (°C) by the dashed black line. (**c**) normalized difference vegetation index (NDVI) is represented by the light green line (On the secondary axis) and river height (cm) by the light blue line.

**Figure 2 ijerph-17-03982-f002:**
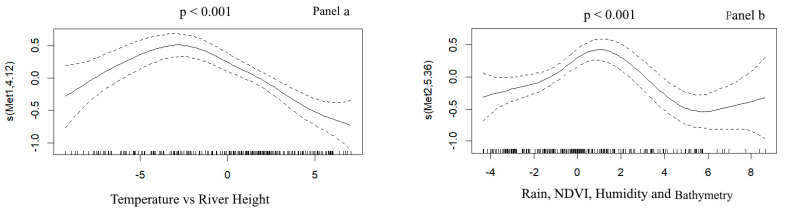
Multivariate generalized additive model (GAM) analysis of synthetic indicators. Panel (**a**) SI 2—temperature vs river height. Panel (**b**) SI 2—rainfall, relative humidity, bathymetry, and NDVI.

**Figure 3 ijerph-17-03982-f003:**
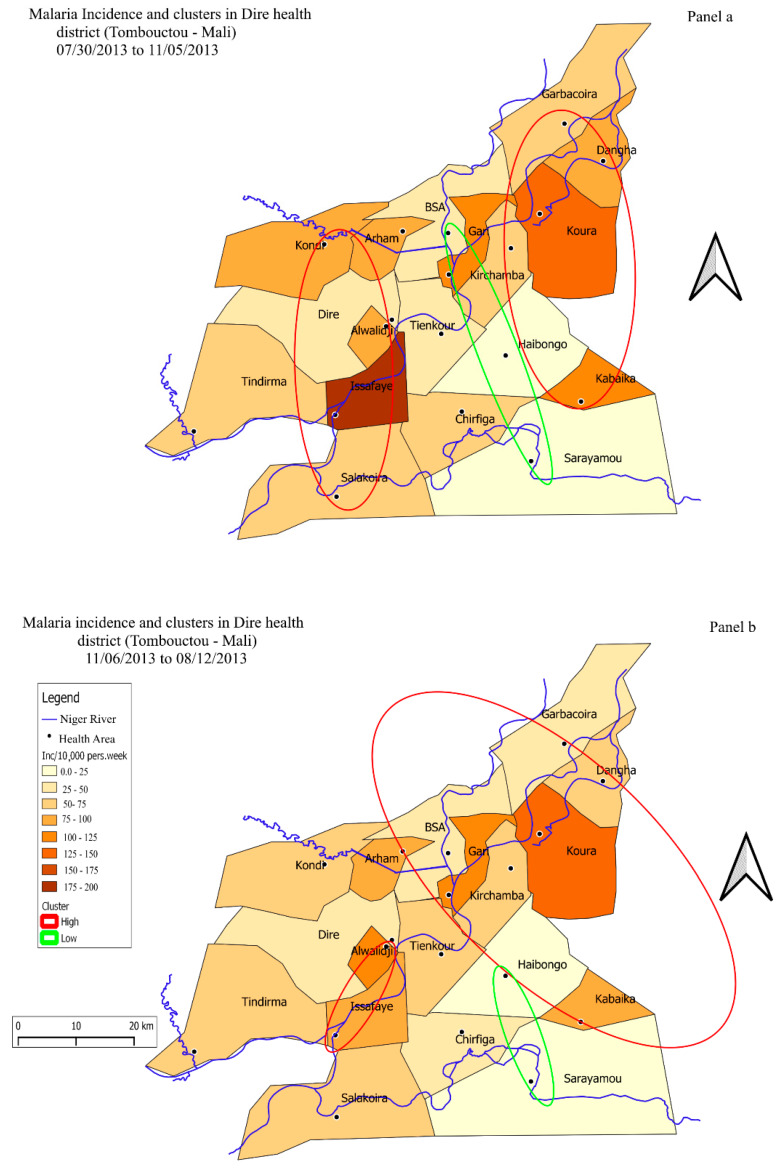
Map of malaria incidence and clusters (hotspots and coldspots) per health area. Panel (**a**): Rainy season. Panel (**b**): Dry season.

**Figure 4 ijerph-17-03982-f004:**
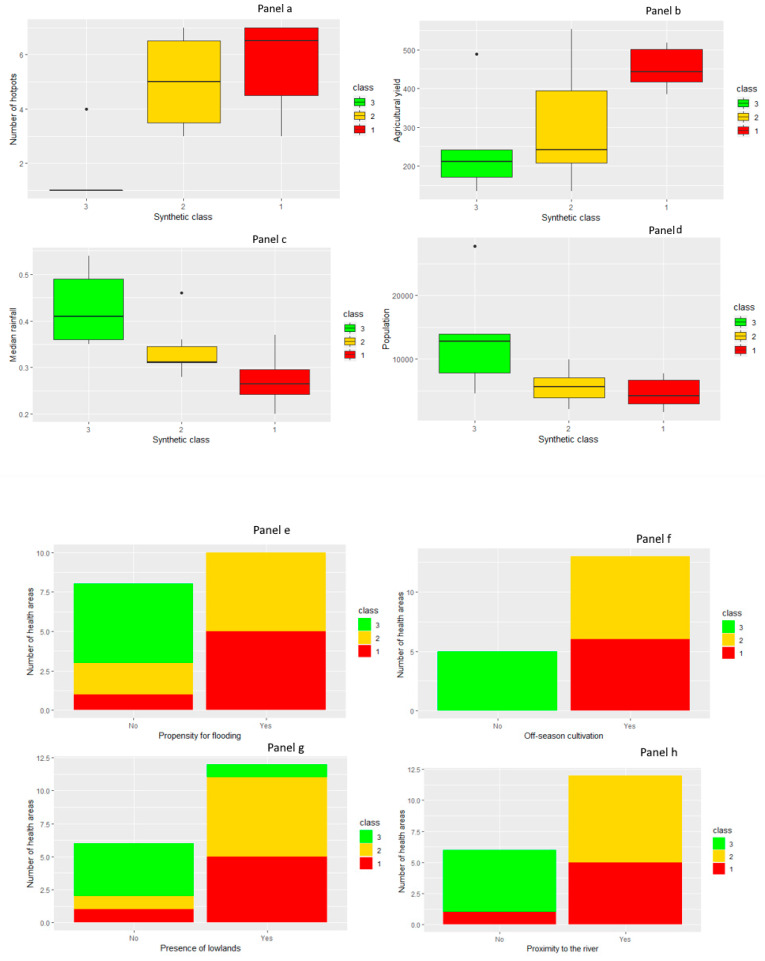
Relationship between risk classes and meteorological, and environmental variables, and risk classes. Panel (**a**) number of hotspot and risk classes, Panel (**b**) agricultural yield and risk classes, Panel (**c**) median rainfall and risk classes, Panel (**d**) population and risk classes; Relationship between environmental and meteorological variables and number of health areas at risk of being a hotspot. Panel (**e**) propensity for flooding and number of health areas at risk of being a hotspot. Panel (**f**) off-season cultivation and number of health areas at risk of being a hotspot. Panel (**g**) presence of lowlands and number of health areas at risk of being a hotspot. Panel (**h**) proximity to the river and number of health areas at risk of being a hotspot. Health areas showing the presence of these variables had a high risk of a being a hotspot.

**Figure 5 ijerph-17-03982-f005:**
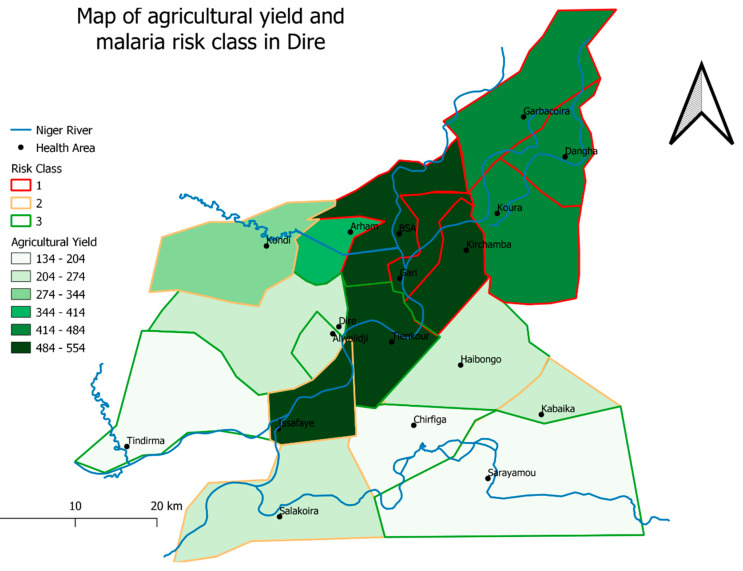
Map of agricultural yield and risk classes in Dire health district.

**Table 1 ijerph-17-03982-t001:** Meteorological and environmental data and malaria incidence by time period.

Date	Number of Weeks	Inc./10,000 Pers.-Weeks	Cumulative Rainfall (mm)	Median Tmax (SD)	Median Tmin (SD)	Median NDVI (SD)	Median Night Relative Humidity (SD)	Median River Height (SD)
30/July/2013	20	43.20	201.4	38.96	23.84	0.24	53.87	207.25
05/Nov/2013				2.56	2.12	0.05	12.34	165.61
06/Nov/2013	31	45.92	0	36.88	16.79	0.17	26.76	266.50
12/Aug/2014				3.93	5.26	0.02	9.20	189.29
13/Aug/2014	22	17.97	218.4	39.08	23.74	0.22	51.42	232.50
18/Nov/2014				2.28	2.44	0.05	9.67	147.17
19/Nov/2014	36	14.38	0	38.28	17.46	0.16	25.29	117.00
28/July/2015				4.71	6.23	0.02	7.10	172.65
29/July/2015	11	49.95	189.6	36.36	23.38	0.29	62.12	262.5
17/Nov/2015				2.20	0.64	0.04	11.13	118.24
18/Nov/2015	38	49.95	0	38.15	17.19	0.17	32.66	359.75
19/July/2016				4.98	6.00	0.04	12.91	180.44
20/July/2016	19	16.69	372.2	37.30	23.22	0.26	58.28	351.50
01/Nov/2016				2.24	1.46	0.05	9.98	185.10
02/Nov/2016	32	19.88	0	38.83	18.12	0.17	26.35	198.50
04/July/2017				4.07	5.89	0.02	6.66	202.66
05/July/2017	19	9.65	217.5	37.81	23.52	0.25	55.95	322.00
07/Nov/2017				1.50	1.95	0.04	9.79	138.29

SD: Standard deviation.

**Table 2 ijerph-17-03982-t002:** Multivariate regression analysis of the link between malaria and risk classes.

Variables	Estimate	Std. Error	IRR	*p-*Value
class 3 (ref)	−0.4298	0.1082		
class 1	0.7950	0.1884	2.21	0.001
class 2	0.5780	0.1264	1.78	0.001

R-squared (adjusted): 0.803.

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
