# Peer review of "Geo-Epidemiology of Malaria at the Health Area Level, Dire Health District, Mali, 2013–2017"

_ijerph, 2020, doi:10.3390/ijerph17113982_

Round 1
Reviewer 1 Report
The manuscript provides a spatial-temporal study about the effects of meteorological and environmental factors on the number of Malaria incidences in Dire health district in Mali. Generally speaking, the research was conducted with logically sound methods, but I am still skeptical about the novelty and impact of the research to be published by IJERPH. Therefore, I recommend a ‘major revision’.
There are several remarks that should be addressed by the authors.
Abstract
Using semicolon (“;”) at the end of every single statement is unusual, so please replace them with dots (“.”).
Introduction
As the study focuses on the geo-epidemiology of Malaria, the current state of the Introduction is insufficient to address the significance and novelty of this study. First, I suggest including Malaria types, most common transmission channels/causes, and current prevention strategies of the government/international organization in Mali. Second, the prevalence, geographic and chronological variation of Malaria infected cases in Mali should also be added. By combining the mentioned information with the clarified objectives of this study, the authors will be able to not only manifest their study’s novelty, but also facilitate readers to understand the current situation in the study site.
Line 43-44: You mentioned “global elimination targets”, what kind of targets are those?
Line 51-52: What does “act as a protective factor in deforested areas” mean? If it means poverty reduction and economic growth promotion, I think it should be replaced by other terms than “protective factor”, because it might result in unnecessary opposition from environmentalists. If it is not, please clarify your meaning.
Line 59-60: you mentioned “land use and land cover has received very little attention”, but I found two very typical studies on environmental factors and Malaria infections in Mali. Please explain the difference/progress between your study and the mentioned studies for addressing the current study’s novelty.
https://malariajournal.biomedcentral.com/articles/10.1186/1475-2875-8-61
https://academic.oup.com/aje/article/163/3/289/59591
This article might also help clarify your paper’s novelty:
https://www.geospatialhealth.net/index.php/gh/article/view/328
Materials and Methods
Please describe the formulas of temporal analysis as well as any other analyses in your study for better validation and reproducibility.
Result
Please provide p-value of Fisher’s exact test and Kruskal-Wallis test and a table of regression results between risk classes and meteorological and environmental factors.
Figure 4 is blurry in the version I received, so please change another picture with higher resolution.
Figure 4 is not mentioned or wrongly indicated in main text.
Conclusion
Please add limitations of the study (e.g. methodologies, data coverage, etc.)
Reviewer 2 Report
This study is well-written, and I believe it addresses an important public health issue. The strengths of the study include
- Strong justification for the need of this study
- The authors provided enough background information.
- Strong statistical analysis
Minor concerns
- The authors stated, “According to WHO estimates” This is the first time WHO is being mentioned so please spell out the full name (World Health Organization) and the subsequent ones can be shortened.
- The authors stated, “In recent decades, however, the global fight against malaria has made a lot of progress with encouraging results, despite disparities between countries” I think it should read “… despite disparities among countries.” Between countries presupposes that you are comparing only two countries. If you are comparing several countries, then I think it should be “among”
- The authors stated “Data on malaria cases (confirmed by rapid diagnostic test)” but it should be “… (confirmed by a rapid diagnostic test or by rapid diagnostic tests)
- The authors stated, “By contrast, older studies found the epidemiology of malaria in Dire to be seasonal and stable over ” The authors should provide references for those studies
- The authors need to acknowledge a limitation or limitations of the study. I can’t think of any limitations now but I believe no study is perfect.
Reviewer 3 Report
Dear all
Greetings
I send u some suggestions (attached document). Congratulations!
Kind regards

Round 2
Reviewer 1 Report
The authors have greatly improved the significance of the manuscript by clarifying the existing gap in the literature. Some places can be improved for better coherence and comprehension.
- In the first paragraph of the Introduction, briefly explaining the aim of the study and what will be done accordingly is necessary. (this suggestion is optional, because it is more like a difference in schools of thought)
- It is more common to present significance level at 5%, 1% and 0.1% as p-value < 0.05, p-value < 0.01, and p-value < 0.001, respectively. Also, please specify in the Method the significance level that you selected.
- There are many typos and grammatical errors as well as formatting errors in the manuscript.
- There are many Figures and Tables in the paper, which will distract the readers, so please consider making them self-contain. (this is also optional)
In sum, some of my aforementioned suggestions are optional, but if the authors seriously adopt them, the clarity and presentation quality would be much enhanced. Good luck!
